# Diabetes and Oral Health (DiabOH): The Perspectives of Primary Healthcare Providers in the Management of Diabetes and Periodontitis in China and Comparison with Those in Australia

**DOI:** 10.3390/healthcare10061032

**Published:** 2022-06-02

**Authors:** Andrew Yun, Yuan Luo, Hanny Calache, Yan Wang, Ivan Darby, Phyllis Lau

**Affiliations:** 1Department of General Practice, The University of Melbourne, Melbourne, VIC 3010, Australia; andrew.yun3@gmail.com; 2Department of Oral and Maxillofacial Surgery, Shanghai Stomatological Hospital & School of Stomatology, Fudan University, Shanghai 200120, China; luoyuan_kq@fudan.edu.cn; 3Shanghai Key Laboratory of Craniomaxillofacial Development and Diseases, Fudan University, Shanghai 200120, China; serene2000@163.com; 4Australian Centre for Integration of Oral Health (ACIOH), School of Nursing & Midwifery, Western Sydney University, Sydney, NSW 2751, Australia; hanny.calache@deakin.edu.au; 5Institute for Health Transformation, School of Health and Social Development, Faculty of Health, Deakin University, Burwood, VIC 3125, Australia; 6Department of Rural Clinical Sciences, La Trobe Rural Health School, La Trobe University, Flora Hill, VIC 3550, Australia; 7Melbourne Dental School, The University of Melbourne, Melbourne, VIC 3010, Australia; idarby@unimelb.edu.au; 8Department of Preventive Dentistry, Shanghai Stomatological Hospital & School of Stomatology, Fudan University, Shanghai 200120, China; 9Department of General Practice, School of Medicine, Western Sydney University, Sydney, NSW 2751, Australia

**Keywords:** interprofessional, collaboration, diabetes, periodontitis, management, perspective

## Abstract

Diabetes and periodontal disease are highly prevalent conditions around the world with a bilateral causative relationship. Research suggests that interprofessional collaboration can improve care delivery and treatment outcomes. However, there continues to be little interprofessional management of these diseases. DiabOH research aims to develop an interprofessional diabetes and oral health care model for primary health care that would be globally applicable. Community medical practitioners (CMPs), community health nurses (CNs), and dentists in Shanghai were recruited to participate in online quantitative surveys. Response data of 76 CMPs, CNs, and dentists was analysed for descriptive statistics and compared with Australian data. Health professionals in China reported that, while screening for diabetes and periodontitis, increasing patient referral and improving interprofessional collaboration would be feasible, these were not within their scope of practice. Oral health screening was rarely conducted by CMPs or CNs, while dentists were not comfortable discussing diabetes with patients. Most participants believed that better collaboration would benefit patients. Chinese professionals concurred that interprofessional collaboration is vital for the improved management of diabetes and periodontitis. These views were similar in Melbourne, except that Shanghai health professionals held increased confidence in managing patients with diabetes and were more welcoming to increased oral health training.

## 1. Introduction

There is a known bilateral relationship between diabetes and periodontal diseases [1,2]. Diabetes is a global epidemic that, whilst previously known as ‘a disease of affluence’, is becoming increasingly common among poorer communities, and is now affecting every country in the world [3]. Periodontitis, on the other hand, is a bacterially induced, chronic inflammatory disease of the supporting tissues of the teeth [4]. It can lead to irreversible tissue loss, tooth loss, and further complications such as malnutrition and lowered quality of life [5]. It is believed that diabetes causes a hyper-inflammatory response to the periodontal micro-biota, and impairs crucial periodontal repair processes [6]. Periodontitis risk is known to be around threefold higher in people with diabetes [7]. Periodontitis can also interfere with glycaemic control and impair diabetes management [7]. Periodontitis is more commonly seen in people with diabetes, while diabetes has a 3–4-fold higher prevalence in people with periodontal disease [8].

Guidelines from the Royal Australian College of General Practitioners recommend that all patients with diabetes should have oral and periodontal health reviews as part of their management [9]. Studies suggest that the use of interprofessional care combined with early screening, prevention, and treatment can improve patient oral health outcomes, ameliorate glycaemic control in people with diabetes and increase efficient delivery of care [10,11].

However, a current lack of collaboration remains between medical and oral health practitioners to manage these diseases, including screening [12,13]. One study indicated that rural general medical practitioners (GPs) in Australia had concerns about their competence to deal with oral health issues when consulting patients with emergency dental problems. Standard practice involved providing short-term pain relief, prescriptions for antibiotics, and recommending patients to see an oral health professional [14]. Another paper looked at barriers within screening in a general practice environment, included legal and reimbursement issues [13]. This is consistent with general medical practice activity data that show that while GP consultations regularly involve patients raising dental concerns, the GPs’ oral health training and awareness are inadequate to deal with complex oral health issues [15].

Integration between primary medical and dental practice needs to be improved to ensure that people with diabetes receive international best practice advice and treatment for their oral and general health needs. Current literature shows that while diabetes is generally managed by a large team of multidisciplinary professionals (including but not limited to general practitioners, nurses, dietitians, diabetes educators, and endocrinologists), oral health professionals are often not part of the team [7,16]. This highlights a significant gap in care for people with diabetes [7].

In 2018, the Diabetes and Oral Health (DiabOH) research team in Melbourne explored the oral health education, interprofessional collaboration and barriers to oral health assessment in primary healthcare professionals [17]. Participants completed online surveys and semi-structured interviews, which were analysed with descriptive statistics and a mixed inductive and deductive approach, respectively. It was found that while participants had a strong determination to collaborate interprofessionally to manage diabetes and oral health, systemic obstacles such as siloed primary healthcare practices and a scarcity of formal interprofessional pathways for referrals existed to hinder this collaborative effort. Furthermore, issues such as time constraints, unintegrated health information systems, and a lack of training regarding the relationship between general and oral health, were found to impede their ability to provide interprofessional care [17].

In Shanghai, the prevalence of diabetes has seen a steady increase over recent years, rising from 9.7% in 2003 to 15.9% in 2016 [18,19]. This increase is seen especially in rural populations as well as younger age groups [18,20]. It is also believed that a large proportion of cases relating to diabetes remain undiagnosed [20].

Smoking is also extremely common in Shanghai, as China remains the world’s largest producer and consumer of tobacco products [21]. This, in combination with diabetes, are significant risk factors for oral health issues that can be detrimental to the incidence and progression of periodontitis [22].

Like Australia, China also lacks interprofessional management of diabetes and periodontitis in the community health setting. Unlike Australia though, China’s oral health workforce has an added barrier layer. There are currently no supportive dental personnel, such as dental therapists, hygienists, oral health therapists, and dental assistants, which can place a burden on running efficient oral health care in China and hinder collaborative opportunities [23].

This paper reports on the replication of the Melbourne study in Shanghai and the findings. We explored the knowledge, experience, and perspectives of Chinese community medical practitioners (CMPs), dentists and community nurses (CPs) on the management of diabetes and periodontitis and on interprofessional collaboration in the primary healthcare setting in China. The findings of this study will be compared with the findings of the Australian study. and, in doing so, this material will inform the development of an interprofessional model of diabetes and periodontal disease management that could be globally implemented.

## 2. Materials and Methods

### 2.1. Ethical Considerations

This project has ethics approval from The University of Melbourne Human Research Ethics Committee (ID 1750825.3) and the Medical Ethics Committee of Shanghai Dental Disease Prevention and Treatment Institute (Batch number: Lukou Fang Lun Shen (2019) No. 011).

### 2.2. Research Design

A quantitative online survey was conducted to assess the self-reported knowledge and confidence of general medical, nursing, and dental professionals in community health services in Shanghai and Australia.

### 2.3. Setting

This study took place at Shanghai Stomatological Hospital and Shanghai Pujin Community Health Service. The health service centre is located on Pujin Street in Pujiang, a town of 102 square kilometres situated approximately 17 km south of central Shanghai [24]. The health centre offers a variety of services including general practice, Chinese medicine, dentistry, and medical imaging, and serves a population of 115,000 resident [25].

The results of this research were compared with those found in the DiabOH Australia study. In that study, participants were recruited from four community health centres in Victoria [17].

### 2.4. Sample and Recruitment

Participants were selected using a purposive approach. Healthcare professionals were recruited from Shanghai Pujin Community Health Service Centre and its affiliated community health practices. The recruitment method utilising WeChat was approved by deans of the Shanghai Stomatological Hospital and Pujin Community Health Service. Researcher LY invited members of the health centre WeChat group to participate in quantitative online surveys regarding their knowledge on the relationship between diabetes and periodontitis, and experience in diabetes and periodontitis management, diabetes screening, periodontitis screening, and interprofessional care. Informed consent was obtained from participants prior to the survey.

### 2.5. Data Collection

The 2018 Melbourne survey questions were translated by researcher LY into Chinese [17]. Translations were checked for accuracy by bilingual researcher PL and piloted by a convenience sample of three healthcare professional colleagues in Shanghai Stomatological Hospital to determine feasibility and resolve any ambiguity regarding the questions.

Two subsequent quantitative online surveys—one 27-item for community doctors and health nurses, and one 28-item for dentists—were then administered through the Survey Star (问卷星) platform, a secure survey management and data entry provider [26]. The community doctors and health nurses’ survey asked participants about their perspectives regarding their past, current, and future oral health training and education, their attention to patient oral health, and their perceived competence in identifying signs and symptoms of oral health issues. Questions also asked about the introduction of an oral health screening tool within diabetes management and their perspectives on collaboration between medical and dental staff, and how this could affect patient outcomes. The dentists’ survey asked about current referral practices, interprofessional care, and management for oral health patients with diabetes. Questions were about their opinion on their scope of practice and their awareness, comfort, and confidence in managing the oral health of their patients with diabetes. Moreover, dentists were asked about the feasibility of using a screening tool similar to the Australian Type 2 Diabetes Risk Assessment Tool (AUSDRISK) in routine practice. It is important to note that currently, no standard risk assessment tool for diabetes exists in China [17].

The surveys were then distributed to participants at Pujin Community Health Service and affiliated community health services via WeChat. Participants were asked to complete their survey in 2 weeks.

### 2.6. Data Analysis

Quantitative data were collected from online surveys and analysed for descriptive statistics using Microsoft Excel (2000). Quantitative descriptive statistics was used to highlight areas of central tendencies and outline the proportion of responses received. This was compared with the survey data from the 2018 Melbourne study conducted in four different community health services in Melbourne.

## 3. Results

In Shanghai, a total of 84 community medical practitioners (CMPs), community health nurses (CNs), dentists, and ‘other’ oral health professionals (OHPs) completed the online survey between 8 and 22 April 2020. The responses of eight ‘other’ OHPs were removed as their professional responsibilities could not be clarified and participants may not have held the relevant clinical experiences necessary for this survey. The responses from 76 CMPs, CNs, and dentists were finally included. Participants’ demographics are listed in Table 1.

Most participants were female between 31 and 40 years old, worked in the public healthcare system, and received their training locally in China. For community doctors and health nurses the typical number of patients seen with these two conditions was 1–10 each week, and nearly all community doctors reported seeing patients with both oral health issues and diabetes risk factors in an average week. In general, community doctors and health nurses saw more patients with type 2 diabetes compared to patients with oral health issues. Dentists commonly saw more than 40 patients with periodontal disease in a typical week. Although most dentists reported that they saw patients with both diabetes and oral health issues each week, fewer patients with both periodontal disease and diabetes or had risk factors associated with type 2 diabetes were seen by dentists compared to CMPs.

Participants were asked about their knowledge on oral conditions and behaviours affecting diabetes management (Table 2). Most participants were aware that poor oral hygiene could impact diabetes management. Over 40% of community doctors and health nurses were not aware that certain diets, alcohol, or smoking had an effect on diabetes management. Community doctors and health nurses seemed to be more aware of oral conditions that can influence diabetes management rather than general lifestyle and health related behaviours. Generally, dentists were more aware of the different conditions that could affect diabetes management compared to community doctors and health nurses. While all dentists were reportedly familiar with gingivitis and periodontitis impacting diabetes management, 27% of community doctors and health nurses were not aware of this link.

Participant responses to questions regarding oral health training, either confidence in diabetes, oral health management, or in combination, referrals, and interprofessional care of diabetes and periodontitis are listed in Table 3. About half of the CMPs and CNs did not feel that oral health was covered thoroughly in their professional training. About half received oral health information from the Internet instead. The majority would welcome the opportunity for continuing oral health education and training.

Although one third of community doctors and health nurses participating were confident in identifying gingivitis and periodontitis, most community doctors were more confident than community health nurses. In general, community health nurses seemed to pay less attention to the oral health of patients compared to community doctors.

Most community doctors and health nurses were aware of the relationship between oral health issues and diabetes and reported that they were confident in managing both conditions. However, most of them were either not comfortable or unsure about talking to patients about oral health. The majority of participants ‘rarely’ or ‘never’ conducted oral health checks for patients with suspected or confirmed diabetes, and reported that they ‘occasionally’ or ‘never’ referred patients to oral health specialists. Most community doctors and health nurses were willing to undertake training in oral health and diabetes management, and agreed that it would be feasible to introduce simple oral health screening for patients with type 2 diabetes. The majority felt that regular oral health screening for patients with diabetes should occur and that they would use a simple tool if available, however one-third (33%) were not sure, or would not, use an oral health screening inspection tool in their practice. 

Most participants agreed that collaboration between general medical and dental practitioners would benefit patients with diabetes and oral health problems, but a proportion (12%) were unsure. Almost all healthcare professionals who participated in this study agreed that there is a role for oral health practitioners to screen for diabetes. 

Nearly all dentists were aware of the link between oral health and type 2 diabetes and the majority responded that they were confident in identifying type 2 diabetes risk factors and managing patients who have both conditions. However, just over half were comfortable discussing diabetes with their patients affected with periodontitis. It was also found that most dentists received their information on conditions that impact oral health treatment and management from professional magazines or journals. All dentists ‘occasionally’, ‘rarely’, or ‘never’ consulted community doctors for further information or referred patients with suspected diabetes, and answered that they should have a role to screen for diabetes in patients with periodontitis. Moreover, all dentists believed that better collaboration with general medical staff would improve patient outcomes. Most dentists were unaware of the Australian Type 2 Diabetes Risk Assessment Tool (AUSDRISK). However, the majority welcomed the introduction of such a tool and either strongly agreed or agreed that this tool would be feasible to be incorporated into the care for patients with periodontal disease. 

There were mixed responses when dentists were asked if the connection between oral health and diabetes was taught well in their oral health course. All dentists were willing to undertake educational training to improve the advice they give to dental patients who also have risk factors for type 2 diabetes, and most welcomed the chance to learn more about the link between oral health and diabetes.

Comparisons of the results from the Melbourne and Shanghai DiabOH studies are found in Figure 1. In both studies, the majority of participants were female except for Chinese dentists, where 62% of the surveyed dentists were male. The majority of OHPs from the DiabOH Melbourne study felt either very comfortable or comfortable discussing diabetes with patients with periodontal disease, while nearly half of dentists in the DiabOH Shanghai study felt not comfortable or not comfortable at all. 

Most OHPs in Melbourne responded that they occasionally, rarely, or never refer their patients to medical professionals, with reportedly only 20% of oral health professionals ‘often’ or ‘always’ referring these patients on. The occurrence of referral was even scarcer in Shanghai, where 100% of dentists responded that they occasionally, rarely, or never opt for referrals. Medical professionals seemed to refer patients the most, where 50% of Australian GPs reportedly ‘always’ or ‘often’ referred patients with periodontal disease to OHPs. Thirty one percent of Chinese community doctors reported facilitating these referrals. For PNs/CNs and OHPs in both studies, the proportion of participants that often referred these patients was always less than 20%. 

When asked if they would welcome continuing training in oral health, 38% of GPs in Melbourne reported that they would welcome further education in oral health, while 80% of community medical practitioners in Shanghai reported that they would gladly receive this opportunity. 

Regarding their confidence in assessing patients with both diabetes and oral health issues, only 18% of GPs and practice nurses in the DiabOH Melbourne study were confident in managing these types of patients, in comparison to the majority (62%) of community doctors and health nurses in Shanghai. 

Nearly all participants in Australia agreed that simple screening procedures for diabetes or oral health conditions were within their professional role. While all dentists in China agreed regarding this screening, 31% of community doctors in Shanghai disagreed or were not sure, while nearly half of the surveyed community health nurses disagreed or were not sure.

All GPs, PNs, and OHPs in the DiabOH Melbourne study agreed that improved collaboration between general medical and oral health professionals would be beneficial for patients with diabetes and oral health conditions. All dentists in the DiabOH Shanghai study approved of this idea, while 7% and 15% of community doctors and health nurses, respectively, were unsure.

## 4. Discussion

The use of a quantitative method allowed us to effectively compare our results with data from the 2018 Melbourne study. The perspectives of professionals from the different fields of medicine, nursing, and oral health provided us with a comprehensive understanding of how these professionals managed their patients with diabetes and periodontitis in practice. Furthermore, this project was the first investigation of its kind in a Chinese community health setting, which will contribute to further study regarding community health clinical practice and interprofessional collaboration. 

Most of the surveyed healthcare participants in Shanghai were female. This is reflective of the fact that most healthcare professionals working in public health services in suburban areas in China are women, although China’s current health workforce reports show that while nurses are overwhelmingly female, 57% of doctors are male [27].

The 2018 Melbourne study similarly had a skew in female representation which was possibly due to the same reason. Another explanation could be that women are more willing to participate in surveys of this nature [28]. Interestingly, a cross-sectional study involving over 500 general practitioners and dentists in Kuwait found that factors significantly associated with having knowledge about the effects of diabetes on periodontal health include an older age, being female, and being a dental professional [29]. 

Our results show that while Chinese dentists seemed to be aware of the connection between oral health and diabetes, the topic could be better taught in medical and nursing training. Our finding is congruent with a 2011 cross-sectional survey study that looked into the knowledge of periodontal conditions related to diabetes in GPs and dentists, which concluded that dentists are significantly more aware of this link, especially when it involves oral conditions, such as gingival bleeding, tooth mobility, and alveolar bone resorption; and that only 50% of participants understood that patients with diabetes were more susceptible to periodontal conditions compared to patients without diabetes [29]. It is clear that the knowledge regarding the link between diabetes and periodontal disease needs to be improved. Through improved training or continued professional development, patients can be provided more effective prevention, treatment, and management of their diabetes and periodontal conditions. 

An unexpected finding in our study shows that the knowledge of community doctors and health nurses in Shanghai regarding lifestyle impacts on diabetes seem to be limited. Their knowledge on this topic area was even less than that of dentists. This lack of awareness was surprising given the links between type 2 diabetes and diet, alcohol, and tobacco use, and the importance of lifestyle management are well-known [6,14,23]. This suggests a need to review medical and nursing diabetes management curricula and continuing professional development in China. 

While dentists in our study seem to possess sufficient knowledge to identify risk factors for diabetes, minimal action was taken to address these for their patients. This presents a significant difference compared to the results from our Melbourne study in 2018, where the majority of surveyed OHPs felt comfortable having this type of discussion [17]. This may potentially be explained by the different ways in which dentistry is taught in China, which focuses on didactic, lectured-based teaching, compared to Western countries that use more problem-based learning and clinical training [30]. Cultural differences can also pose a barrier, as healthcare professionals in China often have a more traditional, paternalistic role that may require less discussion with patients regarding disease management compared to Western doctor’s attitudes that are shifting towards patient-centred communication [31]. 

Most community doctors and health nurses in both this study and DiabOH Melbourne rarely referred their patients to dental professionals. While only 1 in 10 dentist participants in the Melbourne study ‘always’ or ‘often’ referred patients with suspected diabetes to a GP, none of the dentists in this Shanghai study referred [17]. Findings from a study in the United States also found minimal referral that correlated to increased medical comorbidities, where patients often saw at least two general medical providers and experienced oral health problems for more than one year before appropriate referral was made by their healthcare professionals [32]. In China, negligible referral happened between dentists and endocrinologists for periodontitis or diabetes evaluation and management even though the International Diabetes Federation recommends the strengthening of interdisciplinary collaboration to improve general patient outcomes and as a primary means to prevent periodontitis for patients with diabetes [33,34]. Such education can have positive effect not only on patient outcomes, but also improve workplace culture and reduce clinical error rates [35]. 

The lack of referral and collaboration between general medical and oral health care is a global phenomenon, and steps must be taken in order to improve current systems and frameworks. Enhancing interprofessional education between medical, nursing, and dental students may be one way to promote mutual learning and teamwork, improve communication in a collaborative workplace, and improve clinician readiness to collaborate interprofessionally [36,37]. Other recommendations, such as having dedicated personnel assist patients with referrals, improving electronic tools for appointments, and integrating shared electronic health records, have also been made by Atchison et al. to facilitate collaborative care [38]. While participants in our study indicated a desire to increase interprofessional collaboration and improve care for patients with oral-systemic disease, further tangible steps are required to engage healthcare professionals in such practice. 

The self-reported nature of the collected survey data in the DiabOH Shanghai study may have led to recall biases, which may have affected the accuracy of results. The quantitative method did not provide insight into the views, experiences, and attitudes of the survey participants. This anonymous method of collection meant that details regarding occupation or anomalies in results could not be clarified. The predominantly female survey participants may also not be a representative sample of the Chinese healthcare professionals. Additionally, this study was conducted in the public health sector and did not include valuable perspectives from private sector professionals. Future studies should consider conducting either qualitative, feasibility, or in combination, studies to increase understanding and to create a sustainable collaborative model. 

## 5. Conclusions

The DiabOH Shanghai study found that, while Chinese medical, oral health, and nursing professionals believe that increased diabetes and periodontitis screening, increased patient referrals, and improved interprofessional collaboration would be feasible in practice and would provide better disease management and patient care, these objectives were rarely implemented. Community doctors and nurses rarely performed oral health checks for patients with diabetes, and dentists rarely discussed diabetes with patients with periodontitis. 

Some factors that should be considered to improve integrative care involve increasing patient trust in healthcare professionals, enhancing interprofessional education in medical and dental schools, and providing further capacity building opportunities for established professionals. Interprofessional training and collaboration frameworks are required to improve the knowledge, referral pathways, and clinicians’ confidence in the management of diabetes and periodontitis in both Australia and China.

Our findings suggest that further research and understanding of oral and primary healthcare programs in China is needed to facilitate progress and allow for the development of a shared responsibility model in Chinese community health settings. Follow-up interviews are required in order to further explore participants’ perspectives. Subsequently, a tailored oral health screening tool and interprofessional care model should be developed, implemented, evaluated, and optimised. Finally, policy change should occur to promote interprofessional collaboration and enhance the scope of practice of GPs, nurses, and dentists in Australia and China to hopefully improve clinical collaboration and patient health outcomes.

## Figures and Tables

**Figure 1 healthcare-10-01032-f001:**
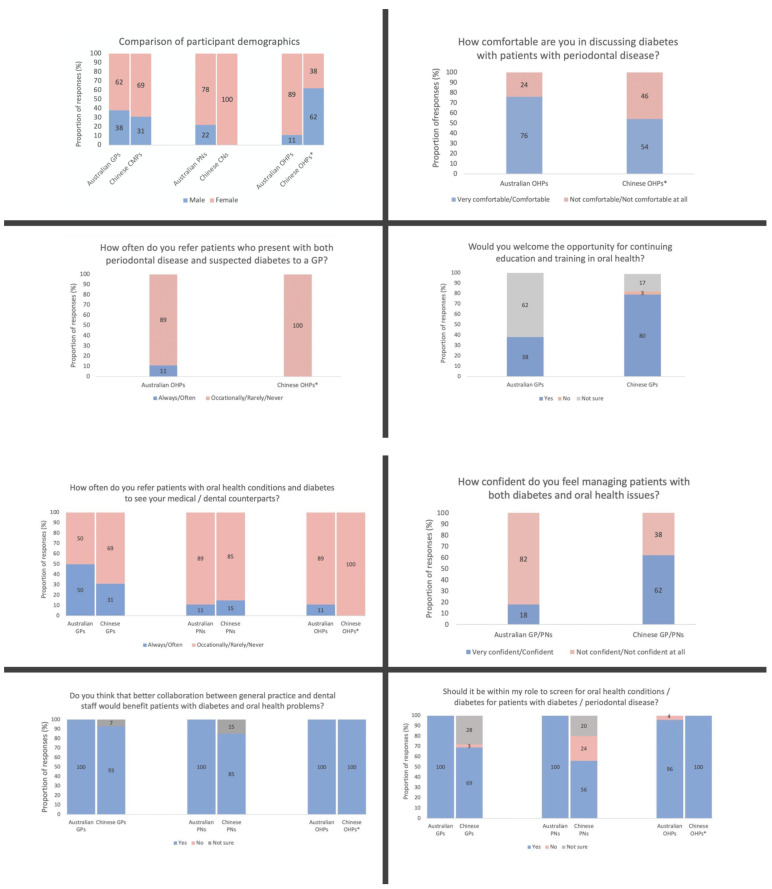
Comparisons between participant responses in this study and those in the Melbourne study [17] (* all OHP participants in Shanghai were dentists).

**Table 1 healthcare-10-01032-t001:** Demographics of participants in Shanghai (community medical practitioners (CMPs), community health nurses (CNs) and dentists).

Variables	CMPs (*n* = 29)	CNs (*n* = 34)	Total CMP/CNs (*n* = 63) No. (%)	Dentists (*n* = 13) No. (%)
**Gender**				
Male	9	0	9 (14%)	8 (62%)
Female	20	34	54 (86%)	5 (38%)
**Age in years**				
<30	4	10	14 (22%)	4 (31%)
31–40	12	19	31 (49%)	5 (38%)
41–50	11	3	14 (22%)	4 (31%)
51–60	2	2	4 (6%)	0
>60	0	0	0	0
**Type of workplace**				
Public	29	34	63 (100%)	13 (100%)
Private	0	0	0	0
Other	0	0	0	0
**Work place location**				
Urban	5	1	6 (10%)	5 (38%)
Suburban	24	33	57 (90%)	8 (62%)
**Hours worked per week**				
0–10	11	12	23 (37%)	0
11–20	2	0	2 (3%)	0
21–25	0	0	1 (2%)	1 (8%)
25–30	2	1	3 (5%)	7 (54%)
>30	14	21	35 (55%)	5 (38%)
**Years of experience**				
0–5	6	7	13 (21%)	6 (46%)
6–10	4	7	11 (17%)	5 (38%)
11–15	6	10	16 (25%)	1 (8%)
16–20	5	5	10 (16%)	1 (8%)
21–25	3	2	5 (8%)	0
26–30	3	1	4 (6%)	0
>30	2	2	4 (6%)	0
**Training received in**				
China	29	33	62 (98%)	13 (100%)
Overseas	0	1	1 (2%)	0
**Number of patients seen per week with oral health conditions**				
0	5	16	21 (33%)	0
1–10	19	9	28 (44%)	3 (23%)
11–20	2	2	4 (6%)	4 (31%)
21–30	1	1	2 (3%)	1 (8%)
31–40	1	2	3 (5%)	0
>40	1	4	5 (8%)	5 (38%)
**Number of patients seen per week with type 2 diabetes**				
0	3	14	17 (27%)	-
1–10	7	9	16 (25%)	-
11–20	2	2	4 (6%)	-
21–30	6	1	7 (11%)	-
31–40	4	2	6 (10%)	-
>40	7	6	13 (21%)	-
**Number of patients seen per week with both type 2 diabetes and oral health issues**				
0	4	14	18 (29%)	2 (15%)
1–10	21	14	35 (56%)	5 (38%)
11–20	1	3	4 (6%)	4 (31%)
21–30	0	1	1 (2%)	0
31–40	2	2	4 (6%)	0
>40	1	0	1 (2%)	0
Not sure	-	-	-	2 (15%)

**Table 2 healthcare-10-01032-t002:** Shanghai participant responses to question regarding knowledge on oral conditions and behaviours affecting diabetes management (community medical practitioners (CMPs), community health nurses (CNs) and dentists).

Condition/Behaviour	Question: Are You Aware of the Following Oral Conditions/Behaviours That Can Impact the Management of Type 2 Diabetes?	CMPs (*n* = 29)	CNs (*n* = 34)	Total CMP/CNs (*n* = 63) No. (%)	Dentists (*n* = 13) No. (%)
**Poor oral hygiene**	Yes	24	32	56 (89%)	12 (92%)
	No	5	2	7 (11%)	0
	Not sure	-	-	-	1 (8%)
**Eating certain foods**	Yes	18	19	37 (59%)	12 (92%)
	No	11	15	26 (41%)	1 (8%)
	Not sure	-	-	-	0
**Drinking alcohol**	Yes	17	16	33 (52%)	10 (77%)
	No	12	18	30 (48%)	2 (15%)
	Not sure	-	-	-	1 (8%)
**Smoking**	Yes	16	19	35 (56%)	9 (69%)
	No	13	15	28 (44%)	1 (8%)
	Not sure	-	-	-	3 (23%)
**Gingivitis**	Yes	22	24	46 (73%)	13 (100%)
	No	7	10	17 (27%)	0
	Not sure	-	-	-	0
**Periodontitis**	Yes	23	23	46 (73%)	13 (100%)
	No	6	11	17 (27%)	0
	Not sure	-	-	-	0
**Dental infection**	Yes	19	24	43 (68%)	12 (92%)
	No	10	10	20 (32%)	0
	Not sure	-	-	-	1 (8%)
**Tooth decay**	Yes	14	18	32 (51%)	-
	No	15	16	31 (49%)	-
**Food trapping**	Yes	14	22	36 (57%)	-
	No	15	12	27 (43%)	-

**Table 3 healthcare-10-01032-t003:** Participant responses to questions regarding oral health training, either confidence in diabetes, oral health, or combined management, referrals, and interprofessional care of diabetes and periodontitis (community medical practitioners (CMPs), community health nurses (CNs) and dentists).

Question	CMPs (*n* = 29)	CNs (*n* = 34)	Total CMP/CNs (*n* = 63) No. (%)	Dentists (*n* = 13) No. (%)
**In your view, did you find the oral health and education components of your medical or nursing course thorough enough?**				
Yes	10	3	13 (21%)	-
No	14	19	33 (52%)	-
Not sure	5	12	17 (28%)	-
**How would you typically find information on oral health?**				
Internet	14	17	31 (49%)	-
Professional magazines or journals	2	3	5 (8%)	-
Colleagues	5	8	13 (21%)	-
Continuing professional education	5	2	7 (11%)	-
Other sources	3	4	7 (11%)	-
**Would you welcome the opportunity for continuing education and training in oral health?**				
Yes	23	28	51 (81%)	-
No	1	3	4 (6%)	-
Not sure	5	3	8 (13%)	-
**How confident are you in identifying the signs and symptoms associated with gingivitis?**				
Confident	16	4	20 (32%)	-
Not confident	2	16	18 (29%)	-
Not sure	11	14	25 (39%)	-
**How confident are you in identifying the signs and symptoms associated with periodontitis?**				
Confident	14	6	20 (32%)	-
Not confident	2	14	16 (25%)	-
Not sure	13	14	27 (43%)	-
**Would you usually consider the oral health of your patients?**				
Yes	11	9	20 (32%)	-
No	3	9	12 (19%)	-
Occasionally	15	16	31 (49%)	-
**Are you aware of the relationship between type 2 diabetes and oral health?**				
Yes	25	25	50 (79%)	-
No	4	9	13 (21%)	-
**How confident do you feel about managing patients with both diabetes and oral health issues (included gum disease)?**				
Very confident/Confident	16	23	39 (62%)	-
Not confident/Not confident at all	13	11	24 (38%)	-
**Are you comfortable talking to patients with diabetes or suspected diabetes about oral health?**				
Yes	10	6	16 (25%)	-
No	7	11	18 (29%)	-
Unsure	12	17	29 (46%)	-
**For patients with diabetes or suspected diabetes, do you give them regular oral health checks?**				
Often	8	6	14 (22%)	-
Occasionally	5	10	15 (24%)	-
Rarely	11	4	15 (24%)	-
Never	5	14	19 (30%)	-
**How often do you refer patients with diabetes and oral health issues to dental specialists?**				
Often	9	5	14 (22%)	-
Occasionally	11	11	22 (35%)	-
Rarely	6	4	10 (16%)	-
Never	3	14	17 (27%)	-
**Do you think that you should screen patients with type 2 diabetes or at risk of diabetes regularly for oral health?**				
Yes	20	19	39 (62%)	-
No	1	8	9 (14%)	-
Not sure	8	7	15 (24%)	-
**If there is an oral health screening tool for GPs and PNs which involves a visual non-invasive inspection with a torch and approximately 5 screening questions, would you use it in practice?**				
Yes	18	25	43 (68%)	-
No	2	1	3 (5%)	-
Not sure	9	8	17 (28%)	-
**Do you agree? It would be feasible in my practice to conduct simple oral health screening (with training) for my patients with type 2 diabetes.**				
Agree	21	28	49 (78%)	-
Disagree	0	1	1 (2%)	-
Not sure	8	5	13 (20%)	-
**Do you agree? The clinical staff in my practice would welcome the introduction of a simple oral health screening tool for patients with type 2 diabetes.**				
Agree	23	29	52 (83%)	-
Disagree	0	0	0	-
Not sure	6	5	11 (17%)	-
**Would you be willing to undertake educational training to assist you in providing oral health advice for your patients type 2 diabetes or risks of diabetes?**				
Yes	26	31	57 (90%)	-
No	3	3	6 (10%)	-
**Do you think there is a role for oral health practitioners to screen for diabetes in their dental patients?**				
Agree	23	21	44 (70%)	-
Partly agree	5	12	17 (27%)	-
Disagree	0	0	0	-
Not sure	1	1	2 (3%)	-
**In your view, would better collaboration between general practice and dental staff benefit patients with type 2 diabetes or risks of diabetes and oral health problems?**				
Agree	27	29	56 (89%)	-
Disagree	0	0	0	-
Not sure	2	5	7 (11%)	-
**Are you aware of the link between oral health and type 2 diabetes?**				
Yes	-	-	-	12 (92%)
No	-	-	-	1 (8%)
**How comfortable are you discussing diabetes with a patient with periodontal disease?**				
Very comfortable	-	-	-	2 (15%)
Comfortable	-	-	-	5 (38%)
Not comfortable	-	-	-	6 (46%)
Not comfortable at all	-	-	-	0
**How confident are you in identifying the risk factors associated with type 2 diabetes?**				
Very confident	-	-	-	3 (23%)
Confident	-	-	-	5 (38%)
Not confident	-	-	-	5 (38%)
Not confident at all	-	-	-	0
**How confident do you feel about managing a patient with both diabetes and oral health issues (including gum disease) in your practice?**				
Very confident	-	-	-	2 (15%)
Confident	-	-	-	6 (46%)
Not confident	-	-	-	5 (38%)
Not confident at all	-	-	-	0
**How do you typically find information on medical conditions that impact on your treatment of oral health?**				
Internet	-	-	-	3 (23%)
Professional magazines or journals	-	-	-	8 (62%)
Colleagues	-	-	-	0
Continuing professional education	-	-	-	2 (15%)
Other sources	-	-	-	0
**How often do you consult with GPs on patients with periodontal conditions and suspected diabetes?**				
Always	-	-	-	0
Often	-	-	-	0
Occasionally	-	-	-	5 (38%)
Rarely	-	-	-	7 (54%)
Never	-	-	-	1 (8%)
**How often do you refer patients who present with both periodontal disease and suspected diabetes to a GP?**				
Always	-	-	-	0
Often	-	-	-	0
Occasionally	-	-	-	8 (62%)
Rarely	-	-	-	4 (31%)
Never	-	-	-	1 (8%)
**Should it be within my role as an OHP to undertake diabetes screening for my patients with periodontal disease?**				
Agree	-	-	-	13 (100%)
Disagree	-	-	-	0
**Do you think there is a role for general practice staff to screen for risk of periodontal disease in their patients with diabetes?**				
Yes	-	-	-	13 (100%)
No	-	-	-	0
Not sure	-	-	-	0
**Would better collaboration between general practice and dental staff would benefit patients with risk factors for type 2 diabetes? (e.g., periodontal disease)**				
Yes	-	-	-	13 (100%)
No	-	-	-	0
Not sure	-	-	-	0
**Are you aware of the AUSDRISK tool for assessing diabetes risk in patients?**				
Yes	-	-	-	1 (8%)
No	-	-	-	12 (92%)
Not sure	-	-	-	0
**Do you agree? It would be feasible in my practice to use the AUSDRISK screening tool (10 questions) for my patients with periodontal disease.**				
Strongly agree	-	-	-	5 (38%)
Agree	-	-	-	7 (54%)
Not sure	-	-	-	1 (8%)
Disagree	-	-	-	0
Strongly disagree	-	-	-	0
**Do you agree? The clinical staff in my practice would welcome the introduction of the AUSDRISK screening tool (10 questions) for my patients with periodontal disease.**				
Strongly agree	-	-	-	4 (31%)
Agree	-	-	-	6 (46%)
Not sure	-	-	-	3 (23%)
Disagree	-	-	-	0
Strongly disagree	-	-	-	0
**In your view, did you receive appropriate education and training in your oral health course regarding the connection between oral health and diabetes?**				
Yes	-	-	-	8 (62%)
No	-	-	-	5 (38%)
**Would you be willing to undertake educational training to assist you in providing advice for your dental patients who also have type 2 diabetes or risk factors for type 2 diabetes (e.g., periodontal disease)?**				
Yes	-	-	-	13 (100%)
No	-	-	-	0
**Would you welcome the opportunity for continuing education and training in the links between oral health and diabetes?**				
Yes	-	-	-	10 (77%)
No	-	-	-	0
Not sure	-	-	-	3 (23%)

## Data Availability

Data will be available on reasonable request to the corresponding author.

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
