# Peer review of "Diabetes and Oral Health (DiabOH): The Perspectives of Primary Healthcare Providers in the Management of Diabetes and Periodontitis in China and Comparison with Those in Australia"

_healthcare, 2022, doi:10.3390/healthcare10061032_

Round 1

Reviewer 1 Report

Dear Authors,

I read this manuscript with a particular and exquisite interest.

It is my pleasure to appraise it and provide recommendations.

In light of my revision, the following corrections can be used by the authors to improve the current version of their manuscript.

With that said, some of the figures can be considered supplemental material.

Consider supporting your introduction section with the following references:

  • https://pubmed.ncbi.nlm.nih.gov/34315460/
  • https://pubmed.ncbi.nlm.nih.gov/32041598/
  • https://pubmed.ncbi.nlm.nih.gov/26242103/
  • https://pubmed.ncbi.nlm.nih.gov/27241250/
  • https://pubmed.ncbi.nlm.nih.gov/34925766/

Other than that, I do not have further comments to share about this well-written submission with the support of a solid expert in the field.

I would be delighted to see the changes in a further revision round.

Thank you.

The Reviewer.

Author Response

Reviewer #1:

Dear Authors,

I read this manuscript with a particular and exquisite interest.

It is my pleasure to appraise it and provide recommendations.

In light of my revision, the following corrections can be used by the authors to improve the current version of their manuscript.

With that said, some of the figures can be considered supplemental material.

Consider supporting your introduction section with the following references:

  • https://pubmed.ncbi.nlm.nih.gov/34315460/
  • https://pubmed.ncbi.nlm.nih.gov/32041598/
  • https://pubmed.ncbi.nlm.nih.gov/26242103/
  • https://pubmed.ncbi.nlm.nih.gov/27241250/
  • https://pubmed.ncbi.nlm.nih.gov/34925766/

Other than that, I do not have further comments to share about this well-written submission with the support of a solid expert in the field.

I would be delighted to see the changes in a further revision round.

Thank you.

We are grateful and thank reviewer 1 for their recommendations and interest in our work.

Thank you for your comments regarding the figures. Our preference is that they stay within the main text, however we would be happy to comply with any suggestions from the editorial team.

Thank you for your suggestions of references. Of these, we have already cited the last one ‘Interprofessional diabetes and oral health management: what do primary healthcare professionals think?.

Of the remainder references, we have now added some information from ‘Attitudes and opinions of Oral healthcare professionals on screening for Type-2 diabetes.’ Additions can be seen below.

‘However, there currently remains a lack of collaboration between medical and oral health practitioners to manage these diseases, including screening

Another paper looked at barriers within screening in a general practice environment, included legal and reimbursement issues.’

The other references seem to be more dental focused, which is not quite relevant to our paper.

Reviewer 2 Report

  1. The main concern of the study is that sample size is relatively small. The number of CMPs and CNs were 63, and the number of dentists was 13. We can only know that the gap between knowledge and clinical practice of periodontitis and diabetes care. But we don’t know exactly whether the gap is significantly or not? Please provide statistically results.
  2. Please provide sample size estimation for participants.
  3. The participants were recruited from one community. The result might not be representative. How do the authors resolve the representative of participants?
  4. It is interesting most dentists know the association between eating certain foods with type 2 diabetes but CMPs and CNs not? How do the authors answer this difference?
  5. Most health care provider know the association between type 2 diabetes and periodontitis, but most of them rarely refer to other specialist. Do the authors provide some opinions or possible resolve for the gap?
  6. Overall, this manuscript is not well-prepared and is not assigned a high enough priority to accept for publication in the present form.

Author Response

Reviewer #2: 

We thank reviewer 2 for their comments and feedback.

The main concern of the study is that sample size is relatively small. The number of CMPs and CNs were 63, and the number of dentists was 13. We can only know that the gap between knowledge and clinical practice of periodontitis and diabetes care. But we don’t know exactly whether the gap is significantly or not? Please provide statistically results.

Our intention was not to analyse the data for statistical significance, our intention was to analyse the data for descriptive statistics and to describe the existing gap.

We are not aiming to prove, but to describe the gaps of knowledge and practice between the different professions.

Please provide sample size estimation for participants.

We did not calculate sample size as mentioned above, our intention was not to analyse the data for statistical significance, our aim was to analyse the data for descriptive statistics in order to describe the current gap.

The participants were recruited from one community. The result might not be representative. How do the authors resolve the representative of participants?

We agree with the reviewer that our findings in this study may not be representative because it came from only one community. Our aim was exploratory to gain an understanding of the experiences and perspectives of different healthcare professionals. The findings will help us to design and construct follow-up surveys or interviews to further increase our understanding.

It is interesting most dentists know the association between eating certain foods with type 2 diabetes but CMPs and CNs not? How do the authors answer this difference?

The reviewer has a great point.

This lack of awareness in non-dental health professionals was unexpected as the links between type 2 diabetes and diet, alcohol and tobacco use are well-known. We have in fact tried to explain this anomaly in lines 319 to 325 in our discussion.

Most health care provider know the association between type 2 diabetes and periodontitis, but most of them rarely refer to other specialist. Do the authors provide some opinions or possible resolve for the gap?

Thank you for your comment. We have written in lines 351 to 361 about this global phenomenon and the recommendations to close the gap, including enhancing interprofessional education, improving workplace communication and improving clinician readiness to collaborate interprofessionally.

In the process of responding to this reviewer, we noted errors in our manuscript and have now made amendments in line 355.

Reviewer 3 Report

  1. need to add numbers to results section in the abstract
  2. any sample size calculation was done?
  3. sampling method?
  4. how the respondents were recruited 
  5. you need to add abbreviation in the table legend 

Author Response

Reviewer #3: 

We thank reviewer 3 for their comments and feedback.

need to add numbers to results section in the abstract

In line 28, we have now added the total survey responses analysed. To conform to the word count requirement of the abstract, we have made necessary minor edits.

any sample size calculation was done?

No, because our intention was not to analyse the data for statistical significance, our intention was to analyse the data for descriptive statistics.

We are not aiming to prove, but to describe the survey responses.

sampling method?

The sampling method is purposive. We have now added that clarification to line 132.

how the respondents were recruited

We have outlined in 2.4 Sample and recruitment how the respondents were recruited in lines 134 to 140.

you need to add abbreviation in the table legend

Thank you very much for this suggestion, we have added the abbreviations in the titles of tables 1 to 3.

Round 2

Reviewer 1 Report

Fortunately enough, the authors’ efforts have paid off. I do not have further recommendations to improve the manuscript.

Reviewer 3 Report

the authors responded well to all my comments